# Examining How Postpartum Videoconferencing Support Sessions Can Facilitate Connections between Parents: A Poststructural and Sociomaterial Analysis

Megan Aston [1,*], Sheri Price [1], Anna MacLeod [2], Kathryn Stone [3], Britney Benoit [4], Phillip Joy [5], Rachel Ollivier [6], Meaghan Sim [7], Josephine Etowa [8], Susan Jack [9], Lenora Marcellus [10] and Damilola Iduye [1]

1  Faculty of Health, School of Nursing, Dalhousie University, Halifax, NS B3H 4R2, Canada; sheri.price@iwk.nshealth.ca (S.P.); damilola.iduye@dal.ca (D.I.)
2  Faculty of Medicine, Dalhousie University, Halifax, NS B3H 4R2, Canada; anna.macleod@dal.ca
3  Department of Human and Social Development, University of Victoria, Victoria, BC V8P 5C2, Canada; kathryn.stone@live.ca
4  Faculty of Science, Rankin School of Nursing, St. Francis Xavier University, Antigonish, NS B2G 2W5, Canada; bbenoit@stfx.ca
5  Department of Applied Human Nutrition, Mount Saint Vincent University, Halifax, NS B3M 2J6, Canada; phillip.joy@msvu.ca
6  BC Women's Hospital & Health Centre, Vancouver, BC V6H 3N1, Canada; rachel.ollivier@dal.ca
7  Research, Innovation and Discovery, Nova Scotia Health, Halifax, NS B3J 0E8, Canada; meaghan.sim@nshealth.ca
8  Faculty of Health Sciences, School of Nursing, University of Ottawa, Ottawa, ON K1H 8M5, Canada; josephine.etowa@uottawa.ca
9  School of Nursing, McMaster University, Hamilton, ON L8S 4L8, Canada; jacksm@mcmaster.ca
10  Department of Human and Social Development, School of Nursing, University of Victoria, Victoria, BC V8P 5C2, Canada; lenoram@uvic.ca
*  Correspondence: megan.aston@dal.ca

**Abstract:** Postpartum support for new parents can normalize experiences, increase confidence, and lead to positive health outcomes. While in-person gatherings may be the preferred choice, not all parents can or want to join parenting groups in person. Online asynchronous chat spaces for parents have increased over the past 10 years, especially during the COVID pandemic, when "online" became the norm. However, synchronous postpartum support groups have not been as accessible. The purpose of our study was to examine how parents experienced postpartum videoconferencing support sessions. Seven one-hour videoconferencing sessions were conducted with 4–8 parents in each group (*n* = 37). Nineteen parents from these groups then participated in semi-structured interviews. Feminist poststructuralism and sociomaterialism were used to guide the research process and analysis. Parents used their agency to actively think about and interact using visual (camera) and audio (microphone) technologies to navigate socially constructed online discourses. Although videoconferencing fostered supportive connections and parents felt less alone and more confident, the participants also expressed a lack of opportunities for individual conversations. Nurses should be aware of the emerging opportunities that connecting online may present. This study was not registered.

**Keywords:** postpartum; videoconferencing; online; support; connecting; feminist poststructuralism; sociomaterialism

## 1. Introduction

The journey of becoming a parent can be a challenging experience, filled with a myriad of changes and emotions, as well as isolation and loneliness [1]. To cope with these stressors, most new parents seek out support and information from a variety of sources, including public health, parenting support groups, community centres, family, and friends [1,2]. Specifically, connecting with other new parents has been identified as an important source

of postpartum support [3], as this offers an avenue for parents to learn from one another, provide validation on what is "normal", and build confidence [4]. In-person postpartum support groups are helpful for new parents. However, some parents may not be able to meet in person due to fatigue, mental health issues, being too busy with a new baby, or a lack of transportation. Online, asynchronous parenting support spaces and groups, such as chat spaces, have slowly been gaining popularity over the last 10 years, and researchers have shown that virtual connections can be supportive and empowering [5–8]. There are fewer examples of synchronous online support sessions and no research studies.

At the beginning of the COVID-19 pandemic (March 2020), public health officials in the province of Nova Scotia, Canada, quickly determined that public health measures were necessary to keep people safe and slow the spread of the virus [9]. This caused the immediate cessation of many in-person programs for new parents, thereby creating an enhanced need to provide programming online. As such, through personal communication, staff from family resource centres, libraries, and parent and baby groups told us that they had provided online support to parents with varying rates of success. No formal evaluations or research had been conducted. Therefore, the aim of this qualitative study was to examine how parents experienced postpartum videoconferencing (specifically using Zoom). We applied the theories of feminist poststructuralism and sociomaterialism to understand how virtual support was experienced by new parents. The findings presented in this article focus on the theme of connecting online. Three themes were too large to fit into one article. The other two themes can be found in two other publications. The first is Zoom etiquette [10], and the second is online postpartum safety [11].

### 1.1. Background

New parents have significant social support needs in the postpartum period, which primarily involve maintaining psychological wellbeing, building the confidence needed to effectively care for their newborns, and knowing when, where, and how to seek help and resources [6,12]. Without social supports, parents are at risk of postpartum isolation. The literature suggests that new parents in the Western world, especially mothers, experience isolation and loneliness in the postpartum period [13], which could lead to serious health challenges, such as postpartum depression [14]. During the COVID-19 pandemic, new parents reported heightened levels of postpartum isolation and depression due to decreased in-person connections and social support [15–19].

Researchers have shown that online supports can provide the opportunity for new parents to connect meaningfully with one another. Through examining an online forum for lesbian mothers with postpartum depression, Alang and Fomotar (5) found that the forum provided significant emotional support. The participants in this study were able to write honestly about their mental health challenges and were supported through validation, advice, and reassurance that they were not alone in their experiences. Similarly, a study exploring a Facebook group for African American breastfeeding mothers reported how the page facilitated the sharing of experiences, ongoing support for one another, improved confidence with public breastfeeding, and prolonged goals for breastfeeding duration [7]. Additionally, researchers conducted an analysis of a Nova Scotia-based online discussion board for first-time mothers, concluding that the space provided empathy, encouragement, information, socialization, and community development for participants [6]. All virtual postpartum support studies to date have been focused on asynchronous chat/forum-based spaces with no real-time audio or camera components.

With many schools, businesses, professional settings, and healthcare settings rapidly transitioning to the virtual environment in 2020, studies began to explore synchronous videoconferencing experiences. Researchers have described "Zoom fatigue" as the exhaustion that results from prolonged participation in video conferences; both mirroring oneself as well as staring at a grid of faces can lead to this fatigue [20]. Further research has reported how students appear more reserved on online platforms and might have a heightened self-awareness on camera, therefore using it less [9,21,22]. In terms of the

healthcare system, one study identified challenges in delivering telehealth services, such as building patient trust, confidentiality, privacy in patients' homes, and technological issues [23]. Conversely, the reported benefits of virtual health visits were accessibility, an increase in patient-centred care, and more rapidly delivered care [23,24]. While there have been studies specific to virtual postpartum healthcare delivery, such as virtual home visiting interventions or breastfeeding support [25], there is a lack of literature regarding virtual support groups for parents and infants during the postpartum period.

A common gap across the existing literature is an overall lack of theorizing the active role of technologies in shaping the connections that are mediated through videoconferencing. Far from neutral entities that exist as the background for human connection, the cameras, buttons, screens, and Internet that enable these connections play an important mediating role. They influence not only when and how but also whether people choose to interact with each other [26]. Technologies therefore change the nature and character of interactions [26]. To our knowledge, this is the first study to explore synchronous videoconferencing as a method for virtual postpartum support groups and to actively attune to how these interactions are not only mediated through but also shaped by videoconferencing technologies. Therefore, we chose to combine a feminist poststructural approach and a sociomaterial approach to examine how parents experienced both the social construction and technical aspects of postpartum videoconferencing sessions.

### 1.2. The Study

The aim of this study was to examine web-based videoconferencing (specifically Zoom) as a mechanism to offer postpartum virtual support groups. We sought to explore the following questions: (1) How do parents and healthcare providers come together using tools and online spaces and experience support in videoconferencing environments? and (2) How are these practices personally constructed within the wider processes and social and institutional discourses of new parenthood?

## 2. Theoretical Perspective and Methodology

### 2.1. Theory

The complexities of social and material factors are heightened in a videoconferencing environment, where traditionally embodied interactions are heavily mediated by communication technologies, such as microphones and cameras [26]. We used two theoretical approaches, sociomaterialism and feminist poststructuralism (FPS), to explore the complexity of how new parents' experiences with technology were socially and institutionally constructed [26], as well as how material technology structured interactions, connections, support, and information [26,27]. Combining these theoretical approaches allowed us to deconstruct how learning and support in a group of postpartum parents were interwoven with social and material factors, including the materiality of videoconferencing technologies and tools (such as the mute button, microphone, camera, and chat space) [26]. Using sociomaterialism required us to examine how social, textual, and technological elements were interwoven, attending to both human actors (parents, babies, researchers) and non-human actors (videoconferencing Zoom platform, cameras, mute buttons, etc.) in our research space [26].

Feminist poststructuralism complimented sociomaterialism, as both theories consider the complex social construction of experiences. Using FPS to examine participant beliefs, values, practices, language, meaning, and agency, we were able to gain a deeper understanding of how parents experienced virtual support groups and negotiated relations of power in interactions with people and technology [27–29]. FPS encourages researchers to look for moments of negotiation to understand how certain beliefs, values, and practices are constructed [27]. Power is understood as relational and complex, where individuals have the agency to "choose" how they navigate different socially and institutionally constructed discourses as they interact with virtual platforms and people. The concept of subjectivity provided a particular understanding of not only how participants understood themselves

but also how they perceived themselves in relation to others and the supports they chose. FPS enabled us to challenge the taken-for-granted position of the "parent" or "mother" as a natural or neutral subject (a position that is socially and institutionally constructed through health discourses) and critically analyze the discourses of motherhood or parenthood, the embedded meanings within these discourses [6], and how these discourses impacted postpartum videoconferencing.

### 2.2. Methodology

2.2.1. Recruitment

Participants were recruited across Nova Scotia Canada from October 2021 to April 2022 using an electronic poster that was circulated on the project's social media accounts and website (mumsns.ca), in local baby stores, family resource centres, women's groups, a local lactation clinic, and the Family Newborn Care Unit of a tertiary provincial pediatric hospital. Recruitment posters invited parents (inclusive of mothers, fathers, and guardians) who were caring for an infant 0–12 months of age to contact the research coordinator by email if they were interested in participating in an online postpartum support session for research purposes. The research coordinator explained the study, emailed the consent form, and obtained verbal consent prior to the videoconferencing support session. At the end of each support session, all participants were invited to email the research coordinator if they were interested in a follow-up one-on-one interview about their experience. The first two participants who emailed the research coordinator from each session were selected for an interview.

2.2.2. Inclusion Criteria and Participant Demographics

To participate in this study, participants had to be caring for an infant between 0 and 12 months of age, living in Nova Scotia, able to speak English, and able to connect to Zoom for the support session. Demographics for all 37 participants can be found in Table 1 and demographics for the 19 participants who were interviewed can be found in Table 2.

**Table 1.** Demographic summary of all 37 participants.

| | |
|---|---|
| Age | Ranged from 28 to 41, with average of 32 |
| Race and ethnicity | 2 of African descent, 1 of Latin descent, 1 of Métis descent, 1 South Asian, 1 Mi'kmaq, 31 White or European ancestry |
| Gender | 1 non-binary, 1 preferred not to say, and 35 heterosexual women |
| Sexual orientation | 4 Queer, 33 heterosexual |
| Annual Household Income | Ranged from CAD 80,000–CAD 210,000, with average of CAD 14,181 |

**Table 2.** Demographic summary of 19 participants who were interviewed.

| | |
|---|---|
| Age | Ranged from 28 to 41 |
| Race and ethnicity | 1 South Asian, 1 mixed Black and White ancestry, 1 mixed Latin and White ancestry, and 16 Caucasian |
| Gender | 1 non-binary identified as "parent", 18 heterosexual women identified as "mothers" |
| Sexual orientation | 2 Queer, 17 heterosexual |
| Annual Household Income | Ranged from CAD 80,000 to CAD 200,000. |

2.2.3. Study Setting: Videoconferencing Support Sessions

The goal of the postpartum support sessions was to connect parents in a casual way to discuss aspects of the postpartum period together via the popular videoconferencing

platform of Zoom. The sessions were facilitated by the principal investigator, who is a registered nurse with experience in public health and postpartum care. Parents were invited to participate in one support session each. The research coordinator, who conducted the consent process with each parent prior to the session, was also present at each session to help facilitation. The support sessions were offered between October 2021 and May 2022, and each session had 4 to 8 parents present. Each session started with a round of introductions, followed by discussions that were guided by "Café Conversations", which were conversation starters for new parents that can be found on the website mumsns.ca. Questions in this facilitation guide were broad and designed to let parents steer the conversation toward things that they wanted to discuss. For example, our first question was always, "where do you get your information from?", which led to conversations about Google, social media, friends, and family and how these sources made parents feel. Parents were told that these discussion questions were just a guide, and they were welcome to talk about whatever they felt comfortable sharing. They were also encouraged to use the audio and video tools of Zoom as they wished. All sessions lasted one hour, and all participants stayed for the duration of the session. We were only able to conduct one session for each group of participants. We have applied for another grant and hope to implement a second study with multiple groups and sessions.

*2.3. Data Collection*

Two primary data types were generated to allow us to document and deepen our understanding of parents' experiences of participating in virtual postpartum support sessions: (1) observational data generated through the audio and visual recordings of each virtual session and (2) in-depth, one-on-one interviews with 19 parents who participated in a session and who expressed interested in and consented to an interview. The visual recordings of the sessions allowed us to both document and explore the social and material elements, for example, how people turned their cameras off and on, how they moved around on camera or stayed still, etc. Individual interviews were conducted by phone, lasted 30–60 min in length, were audio-recorded, and were later transcribed verbatim.

During interviews, we asked participants broadly about their experiences of new parenthood, where they had received support (virtual or in person), how they felt about virtual connections overall, and their experiences in our virtual research support sessions. The 19 in-depth semi-structured interviews were key to understanding participants' experiences. We purposefully asked parents to tell us how they participated during the videoconferencing sessions and how they felt about their experiences. This included a focus on technical and material elements, as well as personal interactions. For example, we asked about participants' decision-making processes for muting/unmuting and having their cameras on or off. We also asked how these decisions made them feel to facilitate a deeper conversation about how they were involved with and negotiated their virtual experiences. We asked about their perceptions of group dynamics, their participation and comfort levels in the online space, and how other participants' decisions regarding muting/unmuting and camera/no camera made them feel. We asked specific questions about their choices to use a particular device, such as a phone or laptop, as well as questions like, "how might our session have gone differently if it were in person?" These questions allowed participants to elaborate on the material and social aspects that the virtual environment brought to the experience of postpartum support sessions.

*2.4. Data Analysis*

Interview transcripts were first analyzed by the primary investigator M.A. and research coordinator K.S. and then by two co-investigators S.P. and A.M. The other members of the team were then able to read summaries of the interviews, quotes, and analyses, after which they added to the analyses and aided in theme development. The analysis was conducted using FPS informed by discourse analysis, as well as sociomaterialism [26,27]. These were appropriate methods to use given the content of the interviews, which were

dually focused on the materials (i.e., cameras, laptops, microphones, and cameras) and the socially constructed discourses of parenthood. We sought to understand how personal experiences of virtual support were impacted by social and institutional discourses in an in-depth way. We undertook this process by first using a guide on how to use FPS informed by discourse analysis [27]. The process began with a careful reading of all interview transcripts, followed by identifying important issues discussed by participants. Then, we considered the values, beliefs, and practices that pertained to quotes, followed by describing the social and institutional discourses informing the issue. We wrote about how the discourses related to the parent and their experiences of power relations, unpacking how the discourse impacts the parent, their negotiations, and their agency in the situation. Using this method of analysis allowed us to pay close attention to the way that parents told their stories and facilitated an exploration of relations of power. We then reviewed each transcript again to identify how the technology structured interactions, connections, and postpartum support and information.

Our sociomaterial analysis focused on deliberately attending to, and noting, the material aspects of the virtual environment. For example, we noted the ways in which parents used and talked about the mute button and how a technological tool could structure a conversation, make parents feel a certain way, or give parents the option to act in certain ways. At the end of every transcript, the research coordinator and primary investigator compiled a list of sociomaterial notes, which were then integrated into the final analysis. Taking this approach allowed us to reconcile the multiple human and non-human data points that comprised our context, considering the social and material complexity of each encounter.

## 2.5. Ethical Considerations

As members of the team were from across Canada, we obtained research ethics approval from eight university or health institution Research Ethics Boards in September 2021, including Dalhousie University (#2021-5660); St. Francis Xavier University (#25404); Nova Scotia Health (#1027776); IWK Health (#1027776); Mount Saint Vincent University (#2021-021); University of Ottawa (#H-10-21-7493); McMaster University (#14334); and the University of Victoria (#21-0456-01). All participants provided verbal consent over the phone to our research coordinator, as was our approved procedure for informed consent. Before participating, all participants were informed that what they said in the support sessions would be kept confidential by the research team; however, researchers could not guarantee that all participants would keep information confidential, even though participants would be asked to do so. Pseudonyms were assigned to participants to protect their identities, and all identifying information has been removed from quotes.

## 2.6. Rigour

Trustworthiness was established in this study through the principles of credibility, confirmability, dependability, and transferability of qualitative research findings [30,31]. We ensured credibility by collecting and analyzing direct quotes from participants and sharing these in our writing to support our findings. Confirmability was attained by having more than one team member complete the analysis and then collectively compare and discuss their analysis and use of the research methodologies. Dependability was accomplished through an audit trail using notes after all meetings that focused on the research process and analysis. Transferability was attained by developing detailed descriptions of the research findings so that readers can agree with the findings in a way that enables them to use the findings in their own practice or scholarly work.

## 3. Results

Of the 37 participants who participated in the virtual support sessions, 19 individuals consented to complete an in-depth interview. Connecting through videoconferencing was a major theme in our study that built upon two other themes: Zoom etiquette and virtual

safety [10,11]. Participant discussions are summarized under the sub-themes of connecting through the camera and connecting through the microphone.

From a sociomaterial perspective, we examined how the camera and microphone impacted interactions. During this process, we could see how certain discourses of parenting were intertwined with the technical interactions, such as the discourse of loneliness and isolation. This discourse informed how participants "connected" online. Sociomaterialism guided the analysis to connect "humanness and technology" to highlight how online technologies (camera and microphone) were used to connect in very unique ways.

All participants in our study indicated that connecting with other parents during the postpartum period was important. In particular, the socially constructed discourse of postpartum isolation has created a need to connect to address loneliness, lack of confidence, and the feeling of being abnormal [8]. To receive and provide support, participants told us that they needed to be present and engaged, which included being seen and heard, as well as seeing and hearing others. Although connecting online might appear to be similar to in-person interactions, it was evident that the camera and microphone constructed the meaning of connecting in a different way, thereby constructing a unique discourse of postpartum virtual connection.

*3.1. Connecting through the Camera*

Each participant had to decide whether to turn their camera on or off, which, as we learned from the interviews, was a thoughtful process. Most participants had their cameras on for the duration of the session. Some participants had family members care for their babies so that they could attend the session by themselves, be on camera, and stay focused. We observed that the parents would usually sit by themselves and not move around, whereas other participants had their babies with them on their laps, breastfed, or played with/entertained their babies. Some participants walked around, folded laundry, and came in and out of frame. There were different comfort levels and ways of interacting while using the camera, but all described their decision-making process in terms of being seen by others. For example, Samira said they struggled with their decision to be the first one to put on the camera.

> I was debating on it before logging in, but when I logged in, I feel like sometimes people are nervous to turn their camera on if no one else has theirs on, so I kind of decided to turn mine on to encourage other people to as well.—Samira

Samira recognized that the camera might instigate the emotion of feeling anxious for other people and therefore chose to use her agency, take the lead, and turn her camera on to encourage others to feel comfortable in doing so. Compared to in-person postpartum groups, where there is no choice but to be seen, Zoom has created a space where one must make a choice as to whether one wants to be seen or not. Choosing whether to be seen was influenced by personal beliefs and comfort, as well as group norms and other participants, where parents often waited to see if others had their cameras on or off before making their decision. Concern about being the only one or the first one to be visible in the group was a common experience. As Devon noted, "you never want to be the only one [with camera on]...it seems weird to have yours on if everyone else has it off". The meaning associated with being seen on camera was very individual. For example, Maria, another participant, noted, "I'm not comfortable breastfeeding in front of people, so I had my video off". Amelia mentioned being "worried" that facilitators would ask her to come on camera and was relieved when this did not happen, as she believed she looked like "a mess" and subsequently preferred not to use the camera. Similarly, Cameron noted how "it can be fatiguing to always be on video and look like you're actively listening and not seem distracted". Previous experiences in virtual spaces had clearly informed the beliefs of these participants and, ultimately, the meaning of an online social discourse of videoconferencing. There was the possibility of being nervous or fearful, worrying about how one looked, or feeling fatigued. Subsequently, these negative feelings might lead someone to leave their camera off when joining a session. This example demonstrates how Samira had to negotiate

relations of power in an online space. In other words, Samira recognized a dominant Zoom discourse where the majority of people usually turn their cameras off. Samira chose to use their agency and challenge this discourse that perpetuated negative feelings about being online on camera. Using the technology of the camera was more than just simply clicking the button or not. There was a purposeful negotiation and thoughtfulness that required participants to reflect upon their own feelings and emotions before engaging with the camera.

In contrast, leaving cameras on was described by all participants as important to facilitate a sense of presence and connection. One participant noted the following:

> Seeing other moms pick up their baby or get a toy to entertain their baby or whatever, just sort of makes you feel connected in a sort of [. . . ] really simple way but it's kind of nice, it feels like you're part of something which is lovely. It definitely is something that I appreciated that you were able to see other folks, it just made you feel like you were hanging out with other people going through the same stuff.—Martina

Feeling connected was extremely important to all participants. They spoke about how seeing each other online during the session was emotionally impactful and used words such as "nice", "lovely", "happy", "comforting", "validating", and "normal". While being able to see other parents might appear to be a "simple" thing, participants reminded us that it was extremely important to see other parents, as it helped them with their own postpartum experience to feel more confident, normal, and less alone. The camera contributed to creating meaning within an online Zoom discourse that included emotional connections.

Having cameras on enabled parents to see others interacting with their babies, wearing comfortable clothes, smiling, breastfeeding, or doing laundry. This provided a sense of connection, reality, and validation of participants' own experiences. By having a clear view of others in their homes and *not* using backgrounds provided by Zoom, participants could see what others were doing. This window into others' homes also added to a feeling of comfort. For example,

> I was actually happy to see that people were in sweatpants, it was like cool I'm not the only person in sweatpants right now, alright. So you know there is comfort in being reminded that your life does not need to be Instagram and I think when you're not seeing people in person it's easy to sort of forget that sometimes and remember that we're all wearing sweatpants and maybe have or have not showered that day because we've been chasing kids around.—Crystal

Crystal reminded us that seeing parents in sweatpants gave her a sense of normalcy. She was challenging an online dominant discourse of having to present oneself as perfect, which represents a discourse that is often prescribed or emphasized by other online platforms, such as Instagram. By seeing others in sweatpants or chasing kids around the house, Zoom afforded participants the opportunity and comfort of seeing others in a more personal way that may not occur at an in-person meeting. Thus, constructing an online postpartum discourse was potentially "comforting" by normalizing the reality of many new parents. Participants could feel comforted by just seeing them do regular things on camera.

Having cameras on and watching others' body language helped to construct a sense of reality and engagement. For example, one participant noted, "everybody was really engaged cause everybody had their camera on for the majority of the time"—Annelise. Similarly, Chelsea described her engagement:

> I think when your camera is on you feel more present right? You're on your game and you feel present in that group. . . I think it was good [seeing other babies]. It just makes it real, it's just like validation right? And just that other people are in it with you and you can see that.—Chelsea

Chelsea described how the camera facilitated feeling present, as she had to be "on her game". This type of presence was more than just seeing faces, it was relational and real,

which ultimately helped her to feel validated in what she was doing as a parent. Similarly, Cameron noted that the camera "makes it seem more real as opposed to just a voice." If there were no camera, maybe the experience would not feel as "real".

Many participants said that the camera allowed them to see facial expressions and body language that then enabled them to feel connected; for example, as described by one participant,

> How they were relaying their stories was very genuine and authentic and I just felt like I could connect with them by seeing them and seeing their kids in a way that you can't without seeing their expressions and the way of expressing themselves with their body. But also it gives them the opportunity to see that the reactions of other people to some of the things that they were saying and seeing maybe that they're not alone and that we're experiencing similar things... I just think it's more personal in that way.—Annelise

The ability to see other participants created a sense of being genuine and authentic, which facilitated a feeling of connection and contributed to the meaning of an online connections discourse. Most participants believed that one really needed to see other people's facial and body expressions on camera to ensure that what they were saying was authentic because their engagement was visible. Esmée discussed how facial expressions helped with validation: "there was one in particular who you'd say something and she'd nod along or you just see her facial expressions change and that kind of helped the validation of what you are saying". Being able to see people's facial expressions and body language was very important and helped to construct a social discourse of connection that included a sense of reciprocity and presence among participants that led to a more personal way of relating and getting to know one another. Esmée noted further:

> It is so much nicer as you're talking if you can see people nodding, like oh they're nodding, or they're shaking their head cause they're oh no we didn't experience that, whatever it is, like you can still see, you still have so much communication context if people's videos are on, not that everyone has to, but I feel like having it on adds to the experience.

It was important to all participants that they develop connections that were genuinely relational. This was not always attainable through videoconferencing since there was the option to turn off one's camera. Seeing how others were physically reacting to other participants in the group was necessary. As Esmee noted, the video "adds to the experience". Elena also noted that being able to see others' reactions made the exchanges more genuine. She shared that "it was nice to be able to put people's words to their faces and see them and relate to them or not relate to them. It just makes them seem more real." The camera was a crucial part of facilitating and constructing an online discourse of connection by helping participants feel authentic connections and exemplifying how relationships can be supported through technology.

There were a few participants who chose to have their cameras off from the beginning of the session until the end. This was a bit disconcerting for some participants, as they could not as easily feel the presence of other participants. For example, Esmée reflected on how one participant in her group did not have their camera on and only unmuted to speak once or twice, saying, "I just didn't even feel her presence". Esmee believed that the camera could facilitate a feeling of presence by placing the participant who did not use the camera into a space of "non-presence". Again, this supports an important aspect of how the camera was an integral part of constructing an online postpartum discourse that incorporated feelings and presence. The camera was an integral part of relations of power, causing some participants to feel uncomfortable when others had their cameras off. Facilitating connections had different meanings for different participants and subsequently affected how they chose to use the camera and interact with others.

Another participant, Crystal, noted that they felt more connected to "the participants who had their videos on versus the one who didn't have it on the whole time." Some

participants even discussed how the participants who had their cameras off the whole time and did not unmute to speak much contributed to uneasy feelings. For example, "the one [participant] where we couldn't see the screen that it felt like an eavesdropper almost [. . . ] with the screen that was blank, that was a creepy kind of who is actually there"—Devon. Similarly, Samira noted, "I think in some ways seeing people's video makes it a little less creepy, where if it was a black screen it might be a little bit more risky feeling"—Samira. While having the camera on contributed to positive feelings of connection and presence, having the camera off created a different online social discourse of feeling uneasy, creepy, and risky.

The way participants thought about, and interacted with, the camera was connected to the way they felt and how they wanted others to feel. From the moment the postpartum support session began, turning one's camera on and off required participants to think about the meaning of the camera and how they wanted to connect. The camera was part of a complex construction of a postpartum discourse of connection. All participants shared how they used their agency to challenge the social discourse of postpartum isolation and contribute to making the online space a place to meet their postpartum needs. For some, this involved having the camera turned on, and for others, this meant having the camera turned off.

### 3.2. Connecting through the Microphone

All participants discussed the importance of sharing and listening to stories, and many noted how hearing each other's stories made them feel comforted, less alone, listened to, and able to relate to one another. Storytelling was constructed as a key feature of the support group sessions and critical to parents' abilities to connect through shared experiences. As Elena shared, "It's humbling to hear similar experiences because it makes you realize that people are going through the same thing as you, and you can relate to people, and you don't feel alone". Similarly, Maria noted how "it seems to be the exact same thing that I am going through so I find that really comforting that you're not the only one out there". Even those who said that the experiences and stories were different from their own said that they felt less alone.

It was important for all participants to address the socially constructed postpartum discourse of isolation during the videoconferencing session, and sharing stories was one way to do this. However, sharing stories online was more complex than in person. Technology created new ways of interacting that were both challenging and supportive. While sharing stories provided opportunities to feel connected, participants also reminded us that it was the way the stories were shared that had an impact on the way they felt. The microphone became an important material aspect and a sociomaterial facilitator that both helped and hindered storytelling. Listening to someone's experience over Zoom and having the ability to respond was different than reading a story over an asynchronous platform or chat space or in person. The reciprocal conversations were influenced and partly structured by the microphone and, in particular, the mute button, where the consistent use of the mute button impacted how stories were shared and contributed to the construction of the discourse of connection.

We observed (and parents told us) that all participants remained muted while not speaking during the virtual support sessions. This meant that when someone was speaking, they were often the only person unmuted (besides facilitators), which allowed for a clear spotlight and audio for the one person discussing their experiences. Participants said that they kept their microphones muted because they did not want loud noises in their homes (such as their babies crying) to distract others in the group. On Zoom, noises coming through microphones can block the audio of others, which made participants extra cautious about potentially overtaking someone's audio while trying to share. In-person conversations do not have to contend with a common microphone that everyone speaks into, potentially at the same time. All participants were familiar with the way that online videoconferencing worked and the nuances attached to the microphone and mute button.

The mute button contributed to constructing an online discourse of connecting that structured the way participants negotiated speaking. The meaning of communication had been shifted due to how the microphone was used. Participants applied certain meanings to the use of the microphone. In person, people can more easily respond, jump in/interrupt, agree, disagree, share their own stories, and validate the speaker. Recognizing both the difficulty of navigating one's microphone online and the need to create a space for connecting with others required participants in the videoconferencing sessions to use their agency and negotiate different relations of power. They had to challenge these conflicting agendas that had been created through the structures of videoconferencing and choose only to unmute when they had a substantial point to add, which resulted in fewer quick statements of agreement and validation, such as "mm hm", "yes", "me too", etc. For example, Martina noted, "I certainly could have said "oh I agree with that" or "oh my goodness that's been my experience too" probably 50 or 60 times [...], but I chose more to speak when I had something really to add". Martina believed that it was important to interject with quick statements to be supportive but recognized the difficulty of doing this on Zoom and therefore compromised her actions. While small interjections are a common and supportive way to communicate in person, online videoconferencing did not support this very well and appeared to devalue this important form of support and connection. Only a few participants used the chat space or emojis, which could have helped with quick supportive comments.

Several participants reflected on how there was a tendency for everyone to take turns speaking during the call:

> Everyone was on the same mindset of Zoom call etiquette where [...] one of us spoke until almost all or all of us had our opinion or our feelings stated [...] I know we never discussed that, it's just something we all naturally did.—Petra

The use of the mute button influenced the ways in which the videoconference session unfolded. Human interactions with non-human technology structured conversations in a way where many participants felt they had to "take turns". While some participants commented on how the practice of turn taking restricted free-flowing conversation found at in-person groups, others discussed how it was beneficial for them to take turns:

> I'm more of an introverted, shy or quiet person [...] I find I'm thrown out by people who talk a lot more generally sometimes, so it's nice to have that space to really be able to share and have a designated time where I could talk.—Cameron

Personal beliefs about how to communicate and interact with others online were intertwined with the physical use of the mute button. Whether a participant preferred to interject with short comments or wait to take their turn, both required participants to think about how they would use technology to help them be present and connect with others in the session. This is an example of how participants negotiated relations of power that worked through an online discourse of connection.

Many participants discussed how it was not just hearing others' stories but the act of responding to shared stories that was helpful for the discussion. One participant said,

> We started referencing each other [...] there was a few times when people referenced me or I referenced others, and I feel like that also made it more likely for people to talk when it was like oh people are listening to me and are reflecting back what I said or relating. I feel like that kind of did open up a bit more of the communication between the people when it was kind of obvious that we were speaking a similar language and having similar experiences.—Devon

Referring back to the experiences of others was valued by this participant and is an example of how many of the participants believed it was important to engage or connect with other parents in a supportive and meaningful way. Referring to, relating to, or "referencing" others during the virtual platform may also have been facilitated by how participant names appeared on the screen, making it easy for participants to use one

another's names and personalizing the experience, even if someone did not have their camera on. In person, this may have been realized by gesturing, quick statements of agreement, or interruptions not common in the virtual environment. The participants in our study believed in and valued listening and responding as practices that facilitated a discourse of connecting. Participants valued being able to verbally *respond* to others in a way that demonstrated empathy, understanding, and support. It was important to be able to share these feelings so that there could be a genuine feeling of reciprocity and presence. Participants used their agency as they strategically thought about and practised unmuting the mute button and responded to others in a way that contributed to empathy and support.

Some participants specifically said that hearing and speaking with others made them feel better about their own self-doubt as parents. One participant mentioned that she was struggling with feeling as though she were living in a bubble and feeling lost:

> Having the opportunity to hear someone else and that they're going through the same experience [. . .] I remember someone told me that it was just normal, it's okay whatever you're doing its fine, and it's okay not to breastfeed, that made me feel better.—Amelia

This participant received direct support from others in their group about issues surrounding breastfeeding. In fact, Amelia never turned her camera on during the session but was able to be reassured by another group member solely through the microphone. In these instances, the microphone was a conduit that contributed to a discourse of connecting and facilitated postpartum support and storytelling.

## 4. Discussion

The importance of an online social discourse of connecting was evident in what participants told us in the interviews. Positive feelings of connection demonstrated how the parents in this study used their agency to challenge, negotiate, and break down relations of power associated with the social construction of isolation in parenthood. As the participants from this study noted, just seeing other people through Zoom was powerful and an example of how they challenged socially constructed notions of isolation in parenthood. However, many participants also spoke about how the camera and mute button could interfere with connecting. Participants had to think about their engagement on many different levels, including the use of the camera and the microphone, and adjust their interactions accordingly for positive experiences of connecting. Interactions of connecting that might have been practised without hesitation in person required more effort and thought when online to ensure presence and focus on seeing and hearing one another.

The social discourse on postpartum isolation had allowed each of the participants to not only join the group to meet others and be fully present to combat isolation but also find ways to be supportive of others during a postpartum videoconferencing session. The participants in this study used words such as comfort, feeling good, relating, real, personal, authentic, genuine, and togetherness to describe how feeling connected and being present through Zoom technology enabled them to feel less alone and more validated as parents.

The findings from other work indicate challenges in establishing presence online. For example, Weinburg [32] described how, in videoconferenced group therapy, presence is hard to achieve through screen relations, where distractions and the absence of eye contact present issues. Nurses and other professionals facilitating virtual groups must pay attention to how cameras connect to participants' emotions. In particular, allowing access to one's home through the camera is a new, novel way to facilitate connections. While many of our participants mentioned appreciating seeing the homes, babies, and faces of parents, others mentioned not wanting to be on camera and even feeling fatigued by being on camera. This has been documented in other research, where studies indicate that mirror anxiety, the feeling of being physically trapped, hyper gaze from a grid of staring faces, and the cognitive load from producing and interpreting nonverbal cues all lead to Zoom fatigue [20]. While we recognize the importance of giving people the option to use their

camera or not (and our participants said we need to give the option), it is important for facilitators to have their cameras on to set a precedent and to encourage people to at least introduce themselves with their cameras on.

Aligned with our findings on turn taking and the mute button, some studies have noted how engaging with others through videoconferencing can result in a disrupted flow or "rhythm" of conversation [33,34]. One study found that participants on Zoom took fewer but longer turns to speak [34], which is congruent with our participants' comments on only speaking if they had something substantial to add to the conversation. In another study, university instructors reported that since students were asked to mute when not speaking, they had less opportunity to provide spontaneous reactions to situations [35]. Although the classroom setting is different from the postpartum support group context, both contribute to the emerging field of literature on communication and connection in the virtual environment.

Connecting virtually with others required participants to actively participate and use their subjective positions and agency to both support and challenge different discourses of online behaviour. Deconstructing the meaning of postpartum online connections using FPS demonstrated the importance of how participants accepted, challenged, and supported the different understandings of connection. While the meaning of connection includes beliefs and values regarding presence, togetherness, authenticity, comfort, and relating to one another, these took on slightly different meanings when practised through videoconferencing. Purposeful engagement with the microphone and camera and human beliefs, values, and emotions, combined with non-human technology, were successfully negotiated by participants through videoconferencing to facilitate genuine connection and support postpartum. Loneliness and isolation have had negative implications on the health and wellbeing of families; therefore, addressing postpartum isolation is an important priority for the nursing profession. This study demonstrates that postpartum videoconferencing offers parents the opportunity to connect in a unique way that is constructed through how they are virtually engaged and how they use their cameras and microphones.

### 4.1. Strengths and Limitations

This research was the first of its kind to explore videoconferencing as a tool for postpartum support groups. By using critical methodologies to understand both human and non-human elements of virtual postpartum support, we were able to deconstruct the meaning of online connections through a focus on the camera and microphone. This study presents findings that are novel to postpartum care and the vital support needed to help parents transition to parenthood. Through the COVID-19 pandemic, society has recognized that not all support can always be in person, and this work represents the first step in understanding how we can support parents remotely through videoconferencing.

The limitations of this research include the relatively homogeneous sample of participants who took part in this study. Most participants were white, heterosexual, and in middle to high socioeconomic positions; thus, it will be important to expand this research to include more diverse participants. Further, participants needed to have access to a device, Zoom, and Wi-Fi to be able to take part in this study, thus excluding all parents without access to these privileges. As we chose to collect data with a larger sample, we were not able to host multiple sessions with the same parents, which could have implications for online connections and the use of videoconferencing tools.

### 4.2. Recommendations for Further Research

Further research should consider how different videoconferencing tools, such as the mute button and camera, influence postpartum support groups over time. Researchers might explore this through the facilitation of virtual postpartum groups with the same group of parents over a longer period to see if the use of the camera and mute button change over time and to see how parents develop relationships virtually over time. It is necessary to understand how videoconferencing support groups might fit into existing

programs at local hospitals, libraries, family resource centres, and other organizations currently supporting postpartum parents. While our findings are specific to the postpartum period, future research should consider studying how connections are formed in different types of support groups.

*4.3. Implications for Policy and Practice*

1. Virtual postpartum support groups are an important means to address postpartum isolation and loneliness. This can be realized by ensuring that parents can connect in ways that are meaningful and relevant to postpartum issues. Understanding how technology, including the camera and microphone, can impact how participants are present, supported, and engaged will enable nurses to effectively facilitate virtual support for parents.

2. Facilitators might consider how people often go through a thoughtful decision-making process about using their cameras or not. Their decision is influenced by personal beliefs and values, as well as group norms. We recommend that facilitators always have their cameras on and that they ask participants to put theirs on, even for just a minute to introduce themselves and establish their presence. While we recognize the importance of cameras, no one should be forced to use their camera.

3. The cameras can help participants connect, be present, and be engaged. They can also facilitate a more "real" sense of being with others and validate experiences by allowing parents to see the (sometimes messy, non-Instagram-filtered) parent home life. This is novel, as in-person parent groups cannot present participant homes. Seeing others in their home environments was comforting and valued by participants. Facilitators should recognize this new aspect of connection in the postpartum period and encourage people to not feel pressured to look perfect but rather embrace the messiness of new parenthood.

4. Camera-facilitated facial expressions can enable parents to engage in reciprocal and relational sharing. It is important for participants to see parents' reactions to stories shared, sometimes by "nodding along", providing more communication context than audio alone.

5. Participants highly valued hearing each other's stories and being validated by others' words, regardless of camera use; however, facilitators should consider how the mute button can help and hinder audio-based connection. Participants said that they all remained muted while not speaking to ensure that the speaker could be heard. This was effective for ensuring that no one was interrupted and everyone had a turn to speak, but it inhibited participants from making informal comments and quick statements of agreement and validation. Facilitators should encourage small comments/interjections in the chat space, with emojis, through camera gestures, through the microphone if possible, or through other creative ways.

6. Participants value when their stories or comments are referenced by others and stated that it can encourage the use of names, noting that a participant's name can be found beneath their picture or video on the screen. Conversation is structured differently online; thus, facilitators must pay attention to the small ways that people can work on relationship building remotely, such as by using each other's names and referring to one another.

**5. Conclusions**

For a host of reasons, postpartum supports may not always be in person. Virtual supports provide an opportunity to address postpartum isolation and facilitate needed connections with other parents that are constructed in particular ways. For example, the camera and microphone are tools of the virtual environment that contribute to complex constructions of meaning and practices of support and connection in the postpartum period. Being present, engaged, supported, heard, and comforted were some of the ways participants wanted to and were able to connect with others. Nurses and healthcare

providers supporting postpartum families must recognize this new shift in supporting parents and families and be able to facilitate videoconferencing groups with the knowledge of how technology impacts group dynamics and the way people can connect online.

**Author Contributions:** Conceptualization, M.A., S.P. and A.M.; methodology, M.A., S.P. and A.M.; formal analysis, M.A., S.P., A.M., K.S., B.B., P.J., R.O., M.S., J.E., S.J., L.M. and D.I.; writing: M.A., S.P., A.M. and K.S.; writing—review and editing, all authors; supervision, M.A. and S.P.; project administration, M.A. and S.P. All authors have read and agreed to the published version of the manuscript.

**Funding:** This research received funding from the Social Science and Humanities Research Council.

**Institutional Review Board Statement:** We obtained research ethics approval from eight university or health institution Research Ethics Boards, September 2021 including Dalhousie University (#2021-5660); St. Francis Xavier University (#25404); Nova Scotia Health (#1027776); IWK Health (#1027776); Mount Saint Vincent University (#2021-021); University of Ottawa (#H-10-21-7493); McMaster University (#14334); and the University of Victoria (#21-0456-01).

**Informed Consent Statement:** Informed consent was obtained from all subjects involved in the study.

**Data Availability Statement:** Data are contained within the article.

**Public Involvement Statement:** Participants in the study were members of the public living in Nova Scotia Canada. Consent was obtained from all participants to use their personal stories as data. Feminist poststructuralism and sociomaterialism were used throughout the research process to guide the study aim, analysis, discussion, and reporting of findings..

**Guidelines and Standards Statement:** We have included all elements of a rigorous and trustworthy qualitative study. https://www.equator-network.org/.

**Conflicts of Interest:** The authors declare no conflicts of interest. The funders had no role in the design of the study: in the collection, analyses, or interpretation of the data; in the writing of the manuscript; or in the decision to publish results.

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
