# Peer review of "Examining How Postpartum Videoconferencing Support Sessions Can Facilitate Connections between Parents: A Poststructural and Sociomaterial Analysis"

_nursrep, doi:10.3390/nursrep14010009_

Round 1

Reviewer 1 Report

Comments and Suggestions for Authors

The paper explores the impact of video-conferencing support during the postpartum period. The content is well-articulated, and the results are effectively presented. However, I have a few minor comments:

1) Could you provide more details on whether the facilitator of the group received specific training for this type of support? It would be beneficial to understand the role of the facilitator in greater depth. 

2) Were there any instances of conflicts during the conversations that led participants to discontinue this form of support?

3) Could you elaborate on whether there were privacy concerns raised by the participants?

4) It appears that each participant only engaged in one support session. Is there any evidence suggesting that multiple sessions with the same participants yield greater benefits or not?

5) Please review and update the reference style. Consider using either numerical citations or the author and year format.

Author Response

Thank you for your comments. They were very helpful and enabled us to clarify some issues in the paper. Please see attached for our responses to all reviewers.

Reviewer 2 Report

Comments and Suggestions for Authors

There are no tables or visuals in this manuscript which makes it a challenge to digest.  I recommend making a table of participant demographics, a table for quotes, and a table for themes that correspond to the framework you are using.

Comments on the Quality of English Language

English language was fine.  Minor edits needed.

Author Response

(The authors gave the same response as above.)

Reviewer 3 Report

Comments and Suggestions for Authors

The manuscript presents a study examining web-based videoconferencing as a mechanism to offer postpartum virtual support groups. The subject of the paper is actual and of great interest. However, the manuscript faces several substantial issues in its current state that must be addressed before publication. Below, I outline my primary concerns:

1.     When specifying the research goals, the authors point to two issues, but it seems that the presented content focuses on the first one, i.e. how do parents and health providers come together using tools and online spaces and experience support in videoconferencing environments. In the text I find fewer references, analyzes and summaries for the second question - How are these practices personally constructed within processes and social and institutional discourses of new parenthood. The authors mention the feeling of loneliness and isolation of parents in the text, but this is not the subject of analyzes of the statements of the study participants and the conclusions of the study.

2.     In my opinion, the theoretical part needs to be strengthened and the theoretical basis of the conducted research should be firmly and more broadly presented. There is a lack of coherence between the theoretical concepts mentioned by the authors and the way of analyzing the results. Perhaps it would be helpful in the analysis to isolate substantive categories based on concepts, according to which the participants' statements would be organized.

3.     The theoretical framework is located in the section on research methodology instead of in the Introduction or Background – why? The presentation of the theoretical base is intended to lead to the research goal and questions. These, however, were presented earlier.

4.     Did the study participants become parents for the first time? Did this matter when recruiting for the study? And if not, did the number of children, their age and therefore different experiences with parenting differentiate the participants' statements? Why was it decided to interview the first two participants who emailed the research coordinator  after participating in the videoconference? What is the methodological justification for this step?

5.     In the Results section, it is worth introducing subsections (sub-themes) that would discuss subsequent emerging categories of participants' statements, distinguished in relation to the theoretical framework. Currently, statements seem to be quoted quite freely without any reference to theory or clear categories. The method of quoting statements also requires consistency - currently, some of them are presented in a block manner, highlighted from the text, and some appear within the text.

6.     The authors start the discussion again with issues related to loneliness and the sense of isolation, which they mention in various parts of the material, but do not highlight them as the main research problem. Perhaps it is worth analyzing the material in this respect and linking the way of using Zoom (camera, microphone) with the feeling of loneliness and isolation, e.g. determining to what extent participation in a videoconference was a remedy for the feeling of loneliness and isolation. Besides, again more firmly grounded theoretical explanation is needed.

Author Response

(The authors gave the same response as above.)

Round 2

Reviewer 3 Report

Comments and Suggestions for Authors

Thank you for responding to my comments and suggestions.